# Forecasting *Staphylococcus aureus* Infections Using Genome-Wide Association Studies, Machine Learning, and Transcriptomic Approaches

Mohamed Sassi,[a] Julie Bronsard,[a] Gaetan Pascreau,[a] Mathieu Emily,[b] Pierre-Yves Donnio,[a,c] Matthieu Revest,[a,d] Brice Felden,[a†] Thierry Wirth,[e,f] (ID) Yoann Augagneur[a]

[a]Inserm, BRM (Bacterial Regulatory RNAs and Medicine)—UMR_S 1230, Rennes, France
[b]Institut Agro, CNRS, Université de Rennes, IRMAR (Institut de Recherche Mathématique de Rennes)—UMR 6625, Rennes, France
[c]Service de Bactériologie-Hygiène Hospitalière, CHU Rennes, Rennes, France
[d]Service des Maladies Infectieuses et Réanimation Médicale, CHU Rennes, Rennes, France
[e]Institut de Systématique, Evolution, Biodiversité (ISYEB), UMR-CNRS 7205, Muséum National d'Histoire Naturelle, Université Pierre et Marie Curie, Université des Antilles, Ecole Pratique des Hautes Etudes, Sorbonne Universités, Paris, France
[f]École Pratique des Hautes Études (EPHE), PSL University, Paris, France

Thierry Wirth and Yoann Augagneur are co-last authors.

**ABSTRACT** *Staphylococcus aureus* is a major human and animal pathogen, colonizing diverse ecological niches within its hosts. Predicting whether an isolate will infect a specific host and its subsequent clinical fate remains unknown. In this study, we investigated the *S. aureus* pangenome using a curated set of 356 strains, spanning a wide range of hosts, origins, and clinical display and antibiotic resistance profiles. We used genome-wide association study (GWAS) and random forest (RF) algorithms to discriminate strains based on their origins and clinical sources. Here, we show that the presence of *sak* and *scn* can discriminate strains based on their host specificity, while other genes such as *mecA* are often associated with virulent outcomes. Both GWAS and RF indicated the importance of intergenic regions (IGRs) and coding DNA sequence (CDS) but not sRNAs in forecasting an outcome. Additional transcriptomic analyses performed on the most prevalent clonal complex 8 (CC8) clonal types, in media mimicking nasal colonization or bacteremia, indicated three RNAs as potential RNA markers to forecast infection, followed by 30 others that could serve as infection severity predictors. Our report shows that genetic association and transcriptomics are complementary approaches that will be combined in a single analytical framework to improve our understanding of bacterial pathogenesis and ultimately identify potential predictive molecular markers.

**IMPORTANCE** Predicting the outcome of bacterial colonization and infections, based on extensive genomic and transcriptomic data from a given pathogen, would be of substantial help for clinicians in treating and curing patients. In this report, genome-wide association studies and random forest algorithms have defined gene combinations that differentiate human from animal strains, colonization from diseases, and nonsevere from severe diseases, while it revealed the importance of IGRs and CDS, but not small RNAs (sRNAs), in anticipating an outcome. In addition, transcriptomic analyses performed on the most prevalent clonal types, in media mimicking either nasal colonization or bacteremia, revealed significant differences and therefore potent RNA markers. Overall, the use of both genomic and transcriptomic data in a single analytical framework can enhance our understanding of bacterial pathogenesis.

**KEYWORDS** *Staphylococcus aureus*, genomics, GWAS, coding and noncoding regions, random forest, trancriptomics, genetic markers, bacteremia, nasal colonization

Address correspondence to Yoann Augagneur, yoann.augagneur@univ-rennes1.fr, or Thierry Wirth, thierry.wirth@mnhn.fr.

The authors declare no conflict of interest.
†Deceased.

*S*taphylococcus aureus is a widespread bacterium, colonizing around 30% of humans (1) along with domestic and wild animals (2). In humans, the spectrum of *S. aureus* infections ranges from superficial skin infections to life-threatening conditions such as bone and joint infections, endocarditis, and bacteremia (3). The severity of these infections is significant, with up to 50% mortality associated with *S. aureus* bacteremia (4, 5). A key factor in the outcomes of bacterial infections is the virulence of the pathogen, in addition to the host immune status and response. Virulence depends on the bacterial genome arsenal and the efficiency of gene expression regulation in response to the host defenses or antibiotic treatments. *S. aureus* nasal colonization usually plays a key role as a starting point of infections that may lead to considerable outbreaks (6). There is a continuum between colonization and infection (7), although among colonized people, only a minority will develop infections. Comparative genomic studies and genome-wide association studies (GWAS) have focused on predicting *S. aureus* virulence (8), on identifying clonal differences in bacteremia-associated mortality (9), on subtle genetic differences between infective endocarditis and bacteremia strains (10), on the identification of strain-specific metabolic capabilities (11), on phyleoepidemiology (12), or, recently, on defining antimicrobial resistance determinants associated with bacteremia (13, 14). Being able to predict in given colonized individuals if they will contract infection and, if so, anticipating how severe the outcome will be should provide a substantial advance notification to set up the appropriate treatment. When human pathogens switch from colonization to infection, they reprogram their gene expression pattern to respond and adapt to the host defense mechanisms and to the multiple stresses encountered (15). Whereas a large number of toxins, transcription factors, and small RNAs (sRNAs) participate in *S. aureus* virulence (16), a breach in host immunity is fertile ground for staphylococcal infections. On the other hand, the presence of hypervirulent (17) and highly adaptable clones (18) suggests that specific and unknown features contribute to the severity of disease (19). Unraveling the genetic factors and their expression that allow some strains to spread aggressively, whereas others remain asymptomatic, is of substantial interest. The genetic changes accompanying the transition from nasal colonization to bloodstream infection in the same individual have been investigated previously in *S. aureus* (20). Interestingly, very few mutations occur, with half producing premature termination codons among some protein-encoding genes and one involved in the truncation of a virulence gene transcription regulator from the AraC family (21). Recently, bacterial fitness studies in human serum showed an inverse correlation between toxicity and disease severity for isolates implicated in invasive staphylococcal diseases (22). Additionally, whole-genome sequencing (WGS) was used to predict the presence of long-term carriers as outbreak sources (23). Despite many studies on *S. aureus* genomics, very few examined whether *S. aureus* colonization could be differentiated from infection at a genome scale and even fewer at the transcriptome level. The advent of next-generation sequencing (NGS) offers the possibility of detecting nucleic acid determinants (DNAs, mRNAs, or sRNAs) that could discriminate the clinical onset of a strain. Taking advantage of hundreds of high-quality *S. aureus* genomes available from both animal and human isolates around the world, we searched for genetic determinants that could predict host preference and, for human strains, colonization, virulence, and disease phenotypes. Knowing the clinical outcome of each isolate included in our data set, we used genome-wide and transcriptome sequencing (RNA-seq) technologies to identify molecular determinants or approaches that could forecast the fate of a colonization or an infection by *S. aureus*. We performed genome-wide association studies followed by the use of the random forest (RF) algorithm to define gene combinations that could differentiate human from animal strains, colonization from diseases, and nonsevere from severe diseases. We found that intergenic regions (IGRs) and coding DNA sequences (CDSs) were critical for accurate strain classification and should serve as templates for future machine learning (ML) strategies. To test whether our approach could be improved by using clonal complex (CC) matrices, we created a second data set solely composed of the most prevalent clonal complex (CC8). However, the reduction in number of available isolates was detrimental for an accurate prediction of unknown

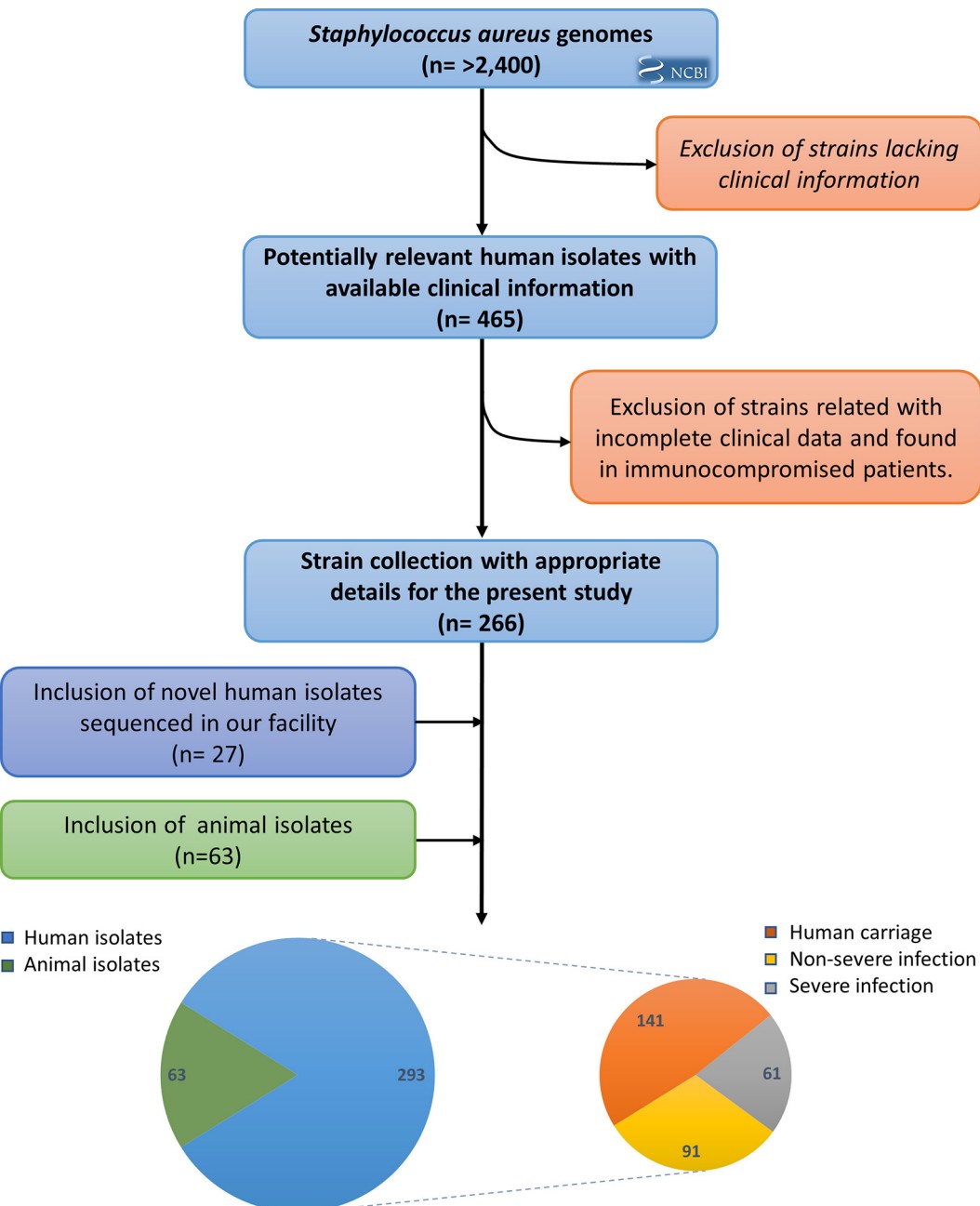

**FIG 1** Overview of the incremental construction of a *Staphylococcus aureus* genome set composed of extensive clinical metadata.

strains. Finally, a transcriptomics approach employed under conditions that mimic transition from colonization to bacteremia enabled the identification of several transcripts, including mRNAs and sRNAs, that are specifically expressed in colonizing or infecting strains. Those transcripts could therefore serve as predictive biomarkers to evaluate disease acuteness.

## RESULTS

**Constitution of a multicountry *Staphylococcus aureus* genome library with clinical information.** To generate an extensive and harmonized collection associated with the metadata, we first performed literature investigations. Fig. 1 illustrates the whole strategy. More than 2,400 *S. aureus* genomes were collected and analyzed. After two independent rounds of reviews devoted to the exclusion of irrelevant studies, 266

Tree scale: 0.01 ⊢——⊣

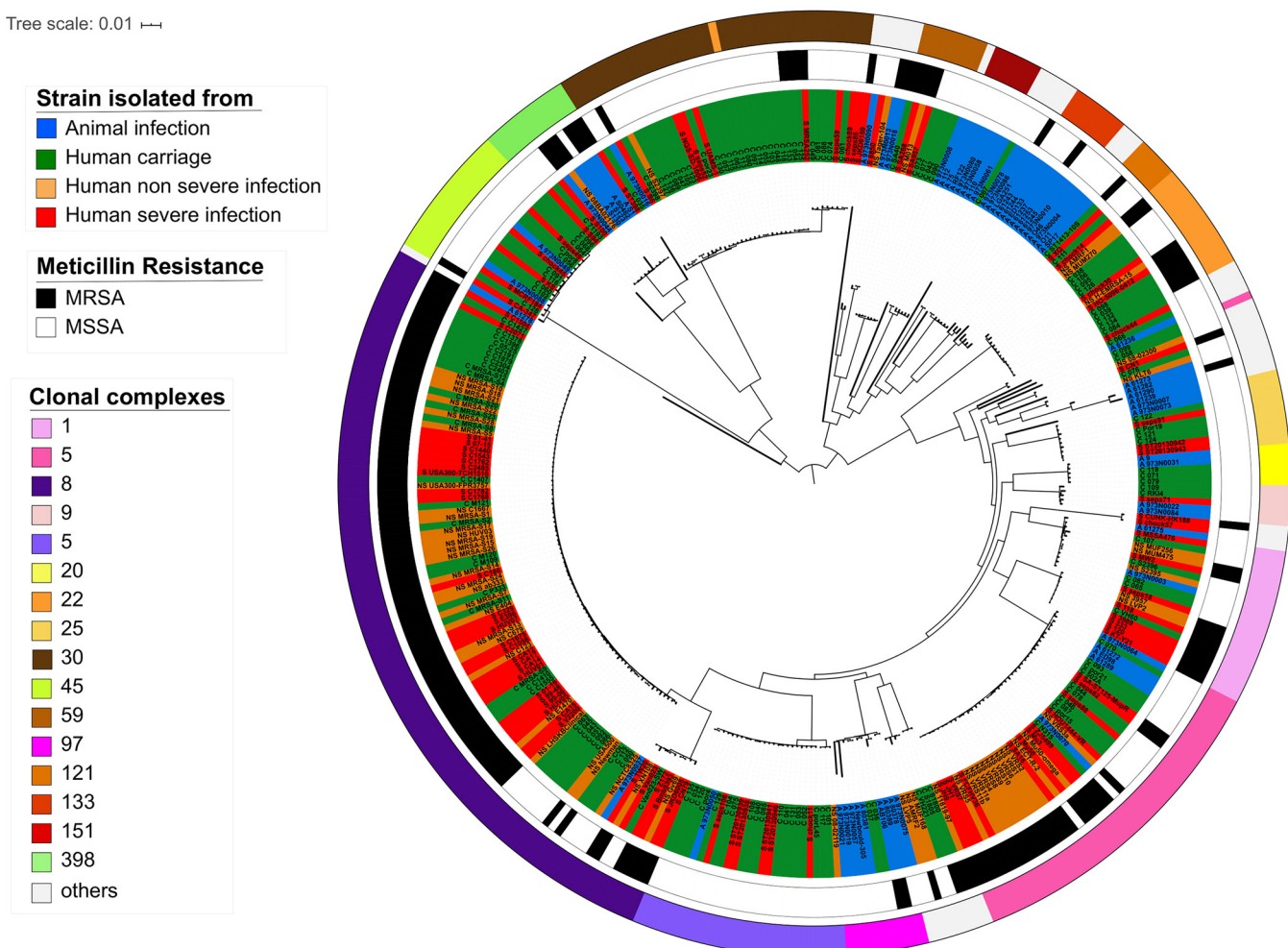

**FIG 2** Phylogenetic reconstruction of multiclonal *Staphylococcus aureus* lineages. Maximum likelihood tree based on 356 genomes. Colors of strain names indicate the type of associated metadata: blue, animal infection; green, human colonization; orange, human nonsevere infection; red, human severe infection. The first outer circle represents the methicillin resistance profile (MRSA for resistance and MSSA for susceptibility). The second outer circle indicates the clonal complex of each strain.

sequenced *S. aureus* genomes from humans with reliable clinical information were conserved. Then, we added 27 new *S. aureus* genomes sequenced in our laboratory and collected from different facilities, as well as 63 genomes sequenced from animal strains, to obtain a final data set of 356 strains (see Table S1 in the supplemental material). Of these, 293 were isolated from human hosts, among which 141 came from colonization and 152 from infections. This last category could be divided into 61 nonsevere and 91 severe infections (Fig. 1). All 356 sequenced strains included in our multicountry study were originally collected from 1943 to 2013 (Fig. S1 and Table S1), except those from Africa, the Middle East, Russia, and Greenland. Most strains (61%) were collected in Europe, with United Kingdom and France accounting for the majority. Eighteen percent of the isolates belong to the North American continent, followed by Latin America (11%), Asia (9%), and Oceania (1%).

We performed a phylogenetic reconstruction based on genome sequences (Fig. 2) and subsequently conducted multilocus sequence typing (MLST) and methicillin-resistant *S. aureus* (MRSA) analyses. MLST analysis predicted 97 different sequence types belonging to 41 clonal complexes (Fig. 2; Fig. S2 and Table S1). In our data set, 27% of the strains belong to clonal complex 8 (CC8), which is one of the most prevalent clonal complexes worldwide both inside and outside health care settings in the United States and Europe (24). The second most prevalent CC was CC5 (Fig. 2; Fig. S2), in agreement with its

worldwide distribution (see Fig. S3 at https://biochpharma.univ-rennes1.fr/supplemental -data) (25). Eleven CCs were exclusively limited to strains isolated from animals ($n = 27$), such as CC133 ($n = 7$), CC151 ($n = 6$), and CC130 ($n = 3$) (Table S1). Based on genome analysis, 43% of the strains were methicillin resistant. The distribution of MRSA and MSSA isolates was not random with respect to CCs. Eighty-three percent of the MRSA isolates were concentrated in only five CCs (CC1, -5, -8, -22, and -30). In those five CCs, approximately half of the strains were MRSA, except for CC8, where the proportion exceeded 80% (Fig. 2; Table S1).

**Pangenomic and genome-wide association study. (i) *S. aureus* genes, sRNAs, and IGR content.** The size of the pangenome, which includes coding sequences, sRNA genes, and intergenic regions (IGRs) of the *S. aureus* species, was derived from the 356 genomes listed in Table S1. First, the CDS analysis performed with Panaroo (26) revealed that from a total of 8,827 predicted groups of orthologs (pan), 1,489 groups of orthologs (54%) belong to the core genome. Within the accessory genome ($n = 7,338$), 2,813 unique genes were identified which corresponded to ~8 singletons per genome (see Fig. S4 at https://biochpharma.univ-rennes1.fr/supplemental-data and Table S2 in the supplemental material). Second, The pan-sRNome (the entire set of sRNAs) was predicted to contain 632 sRNA genes as reported in the SRD database (27), with around 50 of them confirmed experimentally (28). The core of this pan-sRNome contains 271 predicted *srna* genes (42%) (see Fig. S5 at https://biochpharma.univ-rennes1.fr/supplemental-data), indicating that more than half of the predicted sRNA genes are among a variable set of accessory genes, inherited laterally (29). Only one sRNA singleton was identified from multidrug-resistant clone JKD6008 (30). Third, the remaining parts of the genomes are IGRs, which contain regulatory elements. A total of 25,983 independent IGRs constituted our pangenome. Only 137 IGRs were common to all 356 strains (core IGR), suggesting that they contain essential regulatory elements for gene expression. Overall, a total of 8,558 IGRs fell within the accessory genome and 17,288 IGRs were strain specific (see Fig. S6 at https://biochpharma.univ-rennes1.fr/supplemental-data), indicating high variability in those regions.

**(ii) Global functional analysis to investigate differences according to the origin of the strains and their clinical display.** To address whether livestock-, colonization-, or infection-associated (severe and nonsevere) genomes encode determinants specific to their origins, an in-depth functional analysis was conducted on the pangenome. To that aim, each genome was annotated using the COG database (31), and each gene was assigned to functional COG classes and categories. The abundances of genes present in each COG were compared. Significant differences ($P < 0.01$) were found between animal and human isolates for genes encoding the mobilome (i.e., prophages encoding genes and transposons; COG category "X") (see Fig. S7 at https://biochpharma.univ -rennes1.fr/supplemental-data). The genomes of human isolates significantly contain more genes ($P < 0.01$) assigned to the mobilome (mean number of genes, 45) than those of the animal isolates (mean number of genes = 33). Within human isolates, significant differences ($P < 0.01$) were also found within this category between colonization and nonsevere infection isolates. The genomes of nonsevere infection isolates contain more genes from the mobilome (mean number of genes = 56) than colonization isolates (mean number of genes = 43) (see Fig. S8 at https://biochpharma.univ-rennes1.fr/ supplemental-data). Conversely, no significant differences were found between isolates from colonization versus global infection (severe and nonsevere), from colonization versus severe infections, and from nonsevere versus severe infections (see Fig. S8 and S9 at https://biochpharma.univ-rennes1.fr/supplemental-data). Together, these findings show that significant stochastic variation at the mobilome level occurs between livestock and human isolates but also between colonization and nonsevere isolates.

**(iii) GWAS to discriminate strains.** To identify genomic specificities that underline host-associated or severity outcomes, two GWAS approaches were conducted, as was recently done for biofilm-associated genotypes in multidrug-resistant (MDR) *Pseudomonas aeruginosa* (32), but using more restrictive parameters to narrow to small outputs. First, we investigated the presence/absence of CDSs, IGRs, and sRNAs using statistical tools (R and

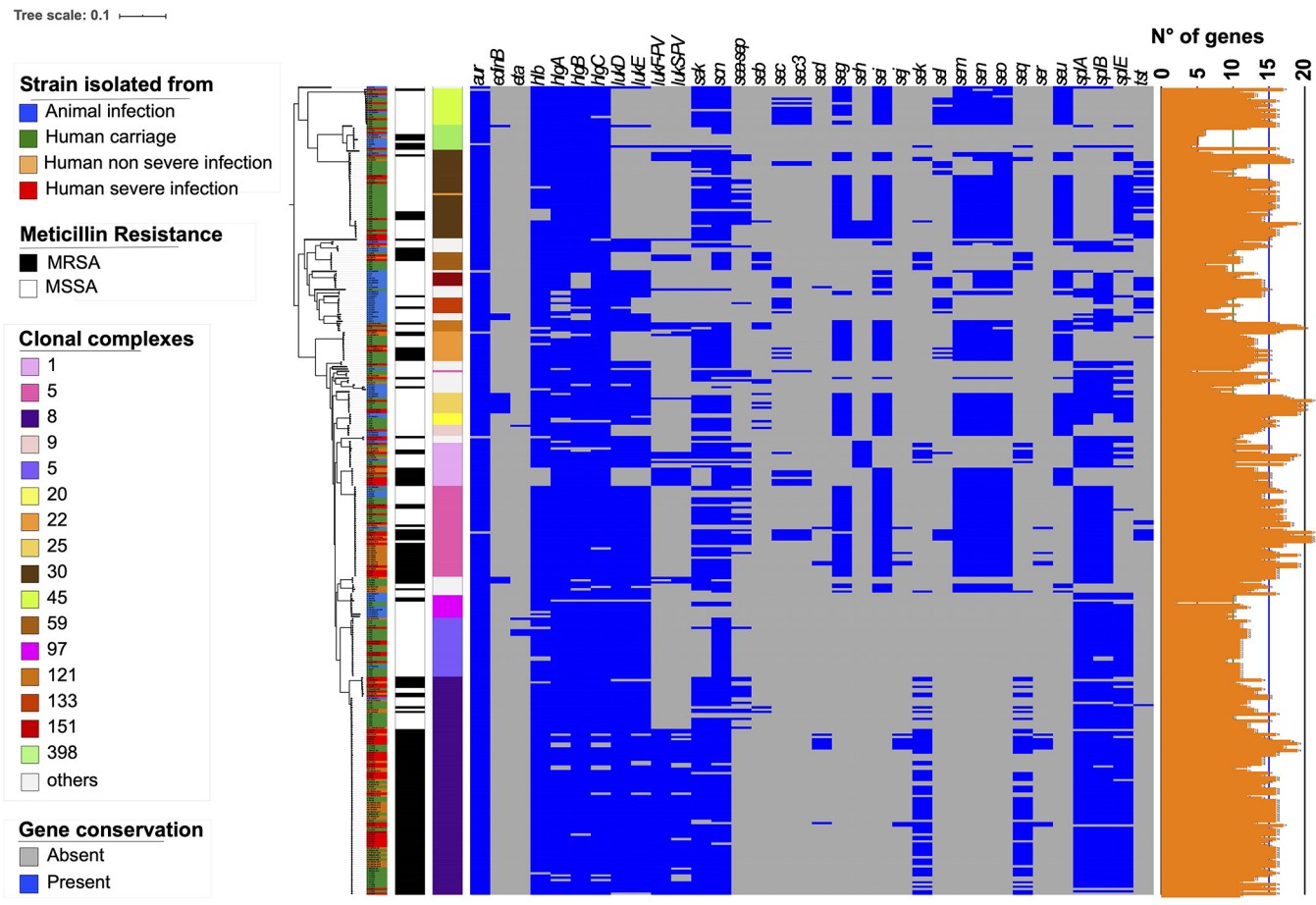

**FIG 3** Distribution of virulence genes within the multiclonal data set. The ML phylogenetic tree is annotated with three colored strips representing the type of isolation of each strain, their methicillin resistance status, and their clonal complex. The binary heat map represents the presence (blue) or absence (gray) of virulence genes identified with the pangenome study. The orange strip on the right indicates the sum of virulence genes present in each isolate.

Scoary) (Table S3). Second, to avoid identifying characteristics of the population structure rather than causative loci that would be truly predictive of bacterial epidemiology, pyseer was utilized (Table S4). Overall, significant differences were retrieved for the CDS and IGR analyses but not for sRNAs, as described below (Tables S3 and S4).

**(iv) Discrimination based on host specificity.** The implementation of a CDS presence/absence matrix with Scoary identified significant differences between animal and human isolates ($n = 48$, Bonferroni's $P < 10^{-5}$), with the virulence genes *sak* and *scn* being among the most significant (Table S3) ($P = 2.85 \times 10^{-35}$ and $2.39 \times 10^{-17}$, respectively). Other genes with strong $P$ values (i.e., $\leq 10^{-12}$) included *recT* (a recombinase gene), group 3780, or group 3965. The second approach with pyseer identified 75 genes significantly associated with human isolates, with *scn* and *sak* being again among the most significant (likelihood ratio test, $P = 3.53 \times 10^{-8}$ and $4.67 \times 10^{-5}$, respectively) and a total of seven genes in common with Scoary (genes coding for 2 transposases and 3 phage proteins besides *scn* and *sak*) (Table S3). Since *sak* and *scn* belong to the virulome, we generated a matrix of the presence/absence of these genes (Fig. 3). This analysis showed that although the presence of both *sak* and *scn* indicates human as a host, a few strains isolated from animals but known to belong to sequence types (STs) able to infect both animals and humans can harbor these two genes (e.g., strain A 61278 on the first line of Fig. 3). This clearly indicates that these two genes are absent in STs that were never reported to infect humans.

Studies of IGRs revealed better consistency between the two GWAS approaches. Scoary identified 32 significant IGRs between animal and human isolates with $P$ values

up to $5.50 \times 10^{-29}$ for cluster 15862, located upstream from a putative nonheme iron containing ferritin that belongs to the ferric uptake regulator-like protein PerR and member of the FUR family (Tables S3 and S5). Other clusters with *P* values of $<1.10^{-15}$ were clusters 15817, 24735, 16230, 11006, and 24115. Among IGRs identified as significant with pyseer, 18/48 were also present in Scoary when livestock strains and human isolates were compared, with clusters 16230, 11006, and 24115 being again recovered but not clusters 15817 and 24735. Cluster 16230 stands for the IGR that corresponds to the promoter region of *sak*, already identified as significant in the CDS analysis (Tables S3 and S5). Cluster 11006 is located between genes encoding a hypothetical protein and a phage major tail protein, whereas cluster 24115 corresponds to the region of integration of prophage Φ13 in NCTC_8325 that truncates a gene encoding a $\beta$-hemolysin (SAOUHSC_02240).

**(v) Discrimination based on clinical outcome.** Strains isolated from humans were further classified based on clinical status. GWAS was also performed to investigate colonization versus infection, colonization versus nonsevere infection, colonization versus severe infection, and nonsevere versus severe genetic traits. For CDS, Scoary identified significant differences in clinical status (Table S3) between colonizations and infections (number of sequences = 15, Bonferroni's $P < 10^{-4}$), colonizations and nonsevere infections (number of sequences = 4, Bonferroni's $P < 10^{-4}$), and colonizations and severe infections (number of sequences = 5, Bonferroni's $P < 10^{-4}$) but not between nonsevere and severe infections. pyseer revealed 6 to 47 genes associated with a clinical category ($P < 10^{-5}$) (Table S4). Twenty-seven genes were associated with strains isolated from an infection versus strains isolated from colonization, with seven being found in common with the Scoary output (Tables S3 and S4). Among genes identified with both GWAS approaches, *mecA* was associated with infection ($P = 2.28 \times 10^{-6}$ and $P = 1.60 \times 10^{-10}$ for pyseer and Scoary, respectively). Similarly, *mecA* was again identified ($P = 7.25 \times 10^{-6}$ and $P = 5.75 \times 10^{-5}$ for pyseer and Scoary, respectively) when colonizations were compared with severe infections.

For IGRs using Scoary, significant differences were found between (Table S3) colonization and infection (number of sequences = 25), colonization and nonsevere infection (number of sequences = 4), and colonization and severe infection (number of sequences = 8) but not between nonsevere and severe infections, as for CDSs. There was again a better consistency between Scoary and pyseer using IGRs, with 20/25 in common between colonization and infection, 1/4 in common between colonization and nonsevere infections, and finally, 8/9 in common between colonization and severe infections. Between colonization and infection, two of the most significant IGRs in common were within tRNA clusters (clusters 25463 and 21327), while cluster 21904 was situated next to the 3' end of *mecA* (Tables S3 and S5). Together, these findings suggest that IGRs might be more critical for differentiating isolates relative to their isolation status and clinical manifestation of disease than other genetic markers/categories.

**(vi) SNP-based discrimination.** We then investigated whether single nucleotide polymorphisms (SNP) were more appropriate for discriminating isolates from colonization and infection and ultimately to be correlated with clinical outputs. For this purpose, a total of 26,577 core genome SNPs, identified in our data set, were examined. Only four significant SNPs (Bonferroni's $P < 0.0001$) were identified using Scoary between animal and human strains, while none were found for the other categories of strains (Table S3). In contrast, pyseer identified a total of 889 SNPs (likelihood ratio test [lrt] $P < 0.0001$) associated with one of the groups of strains (Table S4). One hundred fifty-nine are significantly different between animal- versus human-isolated strains, 66 between colonization versus global infection isolates, 102 between colonization versus nonsevere isolates, 538 between colonization and severe infection, and 24 between nonsevere and severe infections. However, none of the SNPs correlated with *sak*, *scn*, or *mecA*, which were the most discriminating genes using both Scoary and pyseer. Among SNPs identified between colonization and severe infection, the most significant (lrt $P = 4.66 \times 10^{-6}$) was located in *sbnE* (NCTC8325_85880_T_C) (Table S4), which is

involved in siderophore biosynthesis, suggesting that the battle for iron during infection may be a discriminant feature.

**(vii) Homoplastic SNPs, selection, and horizontal gene transfer.** In order to detect whether some SNPs or genes undergo positive selection, and more importantly are associated with a clinical outcome or animal host, we implemented an automated homoplasy analysis with the HomoplasyFinder software (33). In Table S6, we present a list of the 129 most homoplastic SNPs (Homoplasy Index > 0.9). Among them, nucleic acid changes are detected for *csoR*, a gene that regulates copper resistance mechanisms and is often found on plasmids (34), *graR*, which is involved in resistance against cationic antimicrobial peptides and vancomycin (35), and *norB*, which chromosomally encodes an efflux pump whose overexpression can confer MDR to quinolones or tetracyclines (36, 37). Yet, no strong phylogenetic or SNP associations were detected in relation with the goal of this study, underpinning the stochastic nature of local selection, recombination events, and horizontal gene transfers.

**RF as a powerful method to separate strains according to their origins.** The comparative genomics and GWAS methods employed above produced a large amount of data and revealed significant differences between categories that compose our data set. However, they also show some limitations to perfectly distinguish strains based on their environmental origins. Therefore, by combining all these data, we estimated random forest (RF) classification algorithms known to smartly deal with correlation and interaction among features in genomics. RF is a widely used ensemble machine learning (ML) method that aims to aggregate several classification trees using bagging (38, 39). In order to ensure that the estimation of the prediction error rate was unbiased, two-thirds of the data set was used as a training set while, in a second step, the remaining third was used as a testing set. We first took advantage of the out-of-bag error as an estimate of the prediction error based on the training set. Then, we used three different types of genomic matrices as inputs: CDS, IGRs, and CDS plus IGRs (CDS+IGRs). sRNAs were discarded since they were less discriminant after GWAS analyses, which was also confirmed using CART (classification and regression trees) as preliminary inquiries (see supplemental material at https://biochpharma.univ-rennes1.fr/supplemental-data). Trees were generated (i) to distinguish animal from human strains or (ii) to differentiate human isolates based on the clinical status of isolation, namely, colonization versus infections, colonization versus nonsevere infections, colonization versus severe infections, and nonsevere versus severe infections. Overall, the accuracy was high (error rate of the training set, ~10%) in separating strains isolated from animals from strains isolated from humans, reaching 93% (error rate of 7%) when using only IGRs (Fig. 4). However, the error rate ranged from 26% to 33% as a function of the input used for comparisons based on human clinical and carriage status. Although no strong differences were observed regarding the input used, an overall increase of accuracy was measurable when CDS and IGRs were combined. To decipher whether some specific genes or IGRs were critical to build the trees, we took advantage of the variable importance measure of RF and report the five most important genes in Table 1. Interestingly, *scn* was the variable of most importance to segregate animal from human isolates. Group 3413, which was the second most important variable, was also retrieved with the pyseer approach and ranked 15th (Table S4), whereas the three other genes were not previously identified through GWAS. Among a total of 6,638 CDSs used to build trees, group 3780, group 3965, *sak*, and *recT* identified as significant by GWAS were ranked 21st, 35th, 37th, and 54th, respectively. This suggests good correlation between those approaches. Among the 15 CDSs identified with Scoary to discern carriage from infection, the first three (*mecA*, *maoC*, and *ugpQ_1*) were ranked 4th, 5th, and 2nd, respectively. With the IGR matrices, an even better correlation was observed with clusters 15862, 15817, 24735, and 16230 (Table S3), being among the top five variables of importance among a set of 8,045 clusters needed to generate the trees (Table 1; Table S7). Similarly, cluster 25463, identified with pyseer (Table S4), was ranked first and second to distinguish colonization versus infection strains and colonization versus severe infection strains, respectively. To better compare our findings, we extracted the first 20 hits obtained from Scoary, pyseer, and RF measures

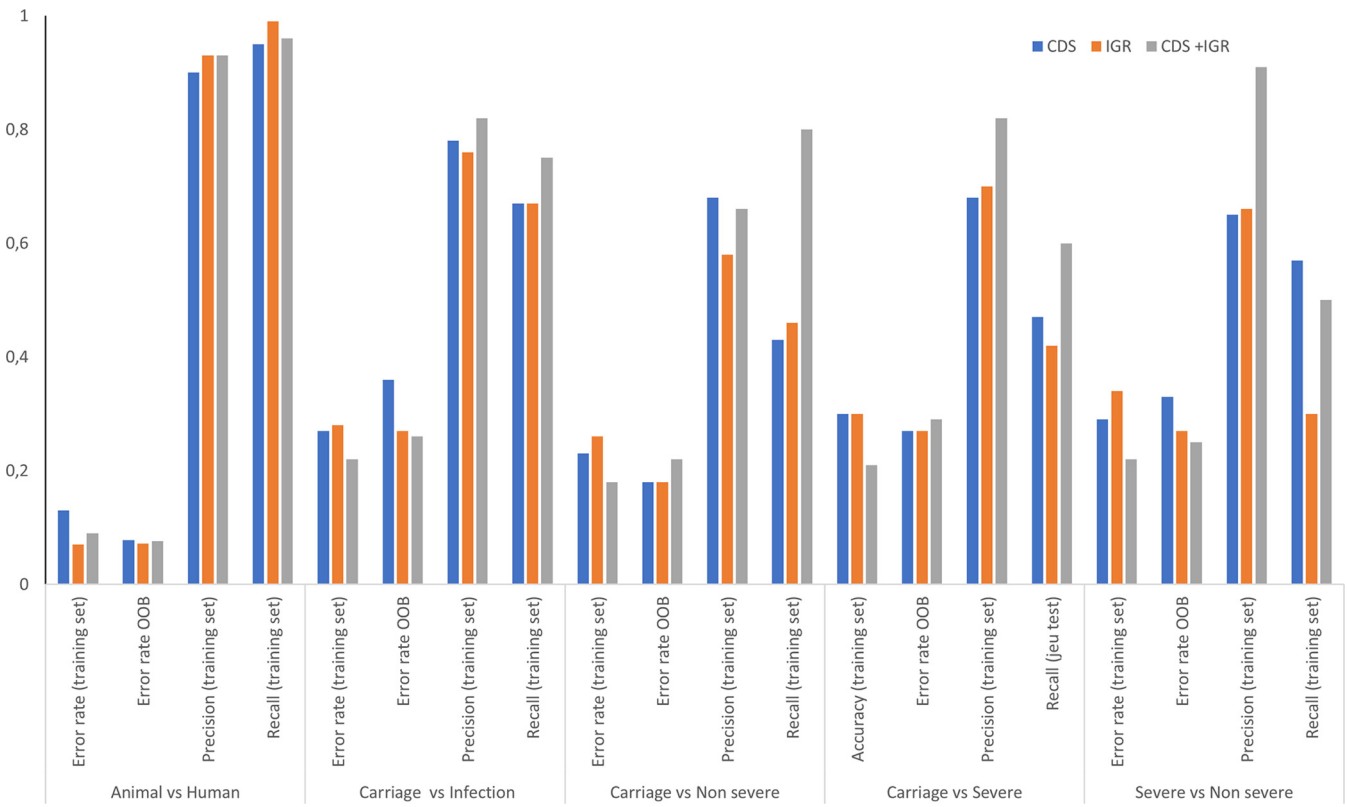

**FIG 4** Host preference or clinical status prediction within the data set of 356 *S. aureus* strains assessed by random forest analysis. OOB, out of bag.

of importance from RF and illustrate the data by using Venn diagrams (Fig. 5). Overall, Fig. 5 shows that better correlation was obtained using IGRs than using CDSs for distinguishing both animal versus human isolates and colonization versus infection. Noteworthy, up to 9 IGRs were in common using the three approaches using IGRs while comparing colonization and infection (Fig. 5C).

To verify the robustness of the RF classification algorithms estimated on the training set, we calculated error rates using the testing set (out of bag). As for the training set, the algorithm was more powerful distinguishing animal strains from human strains than distinguishing strains in the other categories (Fig. 4). An ~7 to 8% error rate was observed to separate these strains using CDS, IGRs, or CDS+IGRs as inputs. Although predictions were less reliable when human isolates were compared based on their clinical status, the error rate did not exceed 18% in discriminating colonization from nonsevere infections when the combination of CDS and IGRs was used while reaching 27% in discriminating colonization versus severe infections, indicating that this comparison is probably the most difficult to make. Also, the use of IGRs or CDS plus IGRs improved the prediction to distinguish colonization versus infection and severe versus nonsevere infections but not for the two other categories. Finally, to verify that errors did not

**TABLE 1** Top five variables of most importance for random forest

| Parameter | Variables of most importance | | |
| --- | --- | --- | --- |
| | **Animal vs human** | **Colonization vs infection** | **Colonization vs severe** |
| CDS | *scn, group_3413, group_1916, lukD, hlb* | *group_5661", ugpQ_1, yezG_2, mecA, maoC* | NA[b] |
| IGR[a] | Cluster_15817, cluster_15862, cluster_24735, cluster_22030, cluster_16230 | Cluster_25463, cluster_21327, cluster_5558, cluster_21209, cluster_1153 | Cluster_21327, cluster_25463, cluster_1153, cluster_19750, cluster_8836 |

[a]Flanking genes are accessible in Table S7 in the supplemental material.
[b]NA, not applicable.

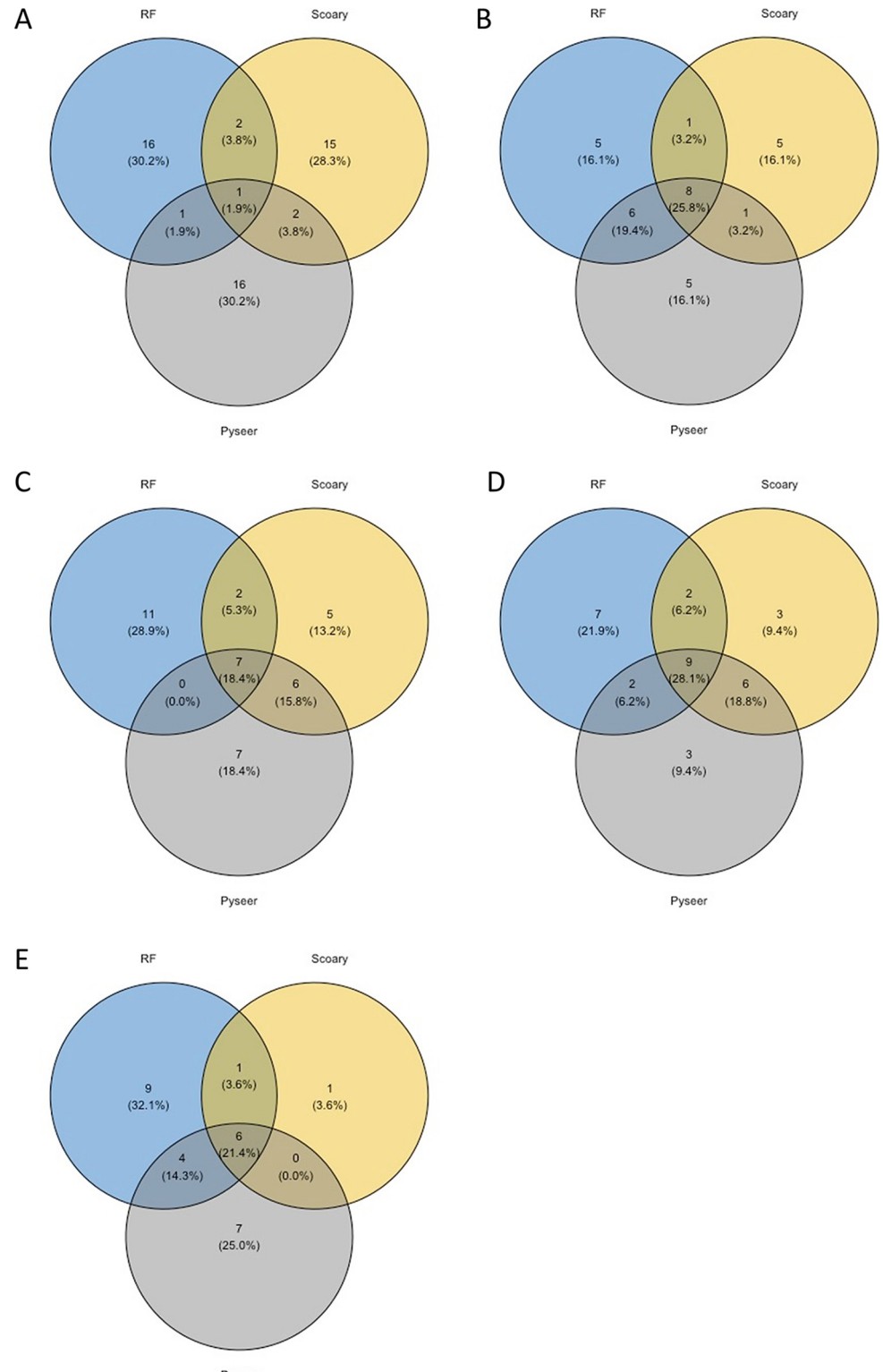

**FIG 5** Comparison and correlation between GWAS and random forest approaches. Venn diagrams were produced after extraction of the first 20 hits obtained from random forest (blue), from Scoary (brown), or from pyseer (purple). (A to E) Comparisons using CDS for animal versus human isolates (A), CDS for colonization versus infection isolates (B), IGR for animal versus human isolates (C), IGR for colonization versus infection isolates (D), and IGR for colonization versus severe infection isolates (E).

**TABLE 2** Use of random forest to predict host preference or clinical status within the ST8

| Matrix | Comparison | Error rate Out of bag (OOB) | Accuracy (training set) | Precision (training set) | Recall (training set) |
|--------|-----------|-----------------------------|-------------------------|--------------------------|------------------------|
| CDS | Colonization vs infection | 0.26 | 0.62 | 0.71 | 0.71 |
| | Colonization vs nonsevere | 0.46 | 0.59 | 0.60 | 0.38 |
| | Colonization vs severe | 0.41 | 0.59 | 0.60 | 0.38 |
| | Severe vs nonsevere | 0.50 | 0.69 | 0.64 | 0.88 |
| IGRs | Colonization vs infection | 0.37 | 0 0.73 | 0.71 | 1 |
| | Colonization vs nonsevere | 0.51 | 0.76 | 0.83 | 0.63 |
| | Colonization vs severe | 0.54 | 0.88 | 1 | 0.75 |
| | Severe vs nonsevere | 0.28 | 0.81 | 0.86 | 0.75 |
| CDS+IGRs | Colonization vs infection | 0.37 | 0.73 | 0.71 | 1 |
| | Colonization vs nonsevere | 0.46 | 0.71 | 0.67 | 0.75 |
| | Colonization vs severe | 0.51 | 0.76 | 1 | 0.5 |
| | Severe vs nonsevere | 0.30 | 0.81 | 0.86 | 0.75 |

cluster in a particular clade, we plotted predicted and actual phenotypes on a phylogenetic tree, which showed that they were mostly random (see Fig. S10 and S11 at https://biochpharma.univ-rennes1.fr/supplemental-data).

Together, these results indicate that the RF methodology is efficient for classifying strains regarding their environmental isolation status and revealed that the use of IGRs is important to improve discrimination. Additionally, the out-of-bag error rates indicate that RF is suitable for a good prediction accuracy in distinguishing animal from human isolates.

**Close-up on ST8 to search for discriminant genomic markers.** We tested whether putting the focus on a more clonal set of strains is relevant to improving the discrimination of isolates based on genomic features. To that aim, we selected ST8 isolates, as they belong to one of the most prevalent clonal complexes (CC8) worldwide and also because we already had 80 ST8 genomes in our main data set. They were classified into three subgroups according to clinical data: colonization, nonsevere infections, and severe infections (Table S1). The 80 ST8 isolates yielded a CDS pangenome size of 4,275 orthologs. The ST8 core genome contains 2,180 CDSs, representing ~79% of the entire genomes, compared with only ~21% when the 356 genomes from the 30 CCs are analyzed (Fig. S2). The accessory genome of the ST8 strains reached a total of 782 CDSs, while 1,313 singletons were found, corresponding to around 16 unique genes per isolate. Gene function annotations (COG databases) revealed that, as for the whole pangenome analysis ($n = 356$), ~43% of the core genome is devoted to metabolism. The ST8 pan-sRNome was composed of 613 sRNA genes. Five hundred six belong to the core genome (80% of those listed in the SRD database [27]), while 92 sRNAs composed the accessory genome and 15 sRNAs were found as singletons (all in JKD6008 strains). Finally, 19 sRNAs reported in the SRD database were missing in all the ST8 isolates. RF was applied to search for genomic markers in the ST8 set (Table 2). Surprisingly, the accuracy fell dramatically, especially when CDSs were used as the sole input. Conversely, validation using the out-of-bag error rates showed that error rates were generally lower after using the training set with CDSs, except for prediction of severe or nonsevere infections, where the use of IGRs substantially improved predictions. These data suggest that the reduction of the set's size is detrimental to forecasting the outcome.

**Use of RNA-seq under conditions mimicking nasal colonization and bacteremia to identify potent transcriptomic markers in ST8. (i) Bacteremia medium versus SNM3 medium.** Although promising, GWAS or RF performed on the multiple-CC data set or on ST8 illustrated some limitations in discriminating or predicting strain origins and phenotypic outcomes. To further widen our approach, we investigated whole-gene expression by RNA-seq. We selected 10 ST8 isolates from our ST8 data set (Fig. 6; Table S1). Four were isolated from asymptomatic carriers and six from patients with severe infections treated in our hospital. Within the severe infection strains, we distinguished strains that yielded bacteremia without organ failure ($n = 3$) from the one that yielded septic shock ($n = 3$) according to the Third International Consensus Definitions for Sepsis and Septic Shock (40). Cells were grown in medium that mimics nasal colonization (SNM3) and then transferred to bacteremia mimicking medium (BMM), which

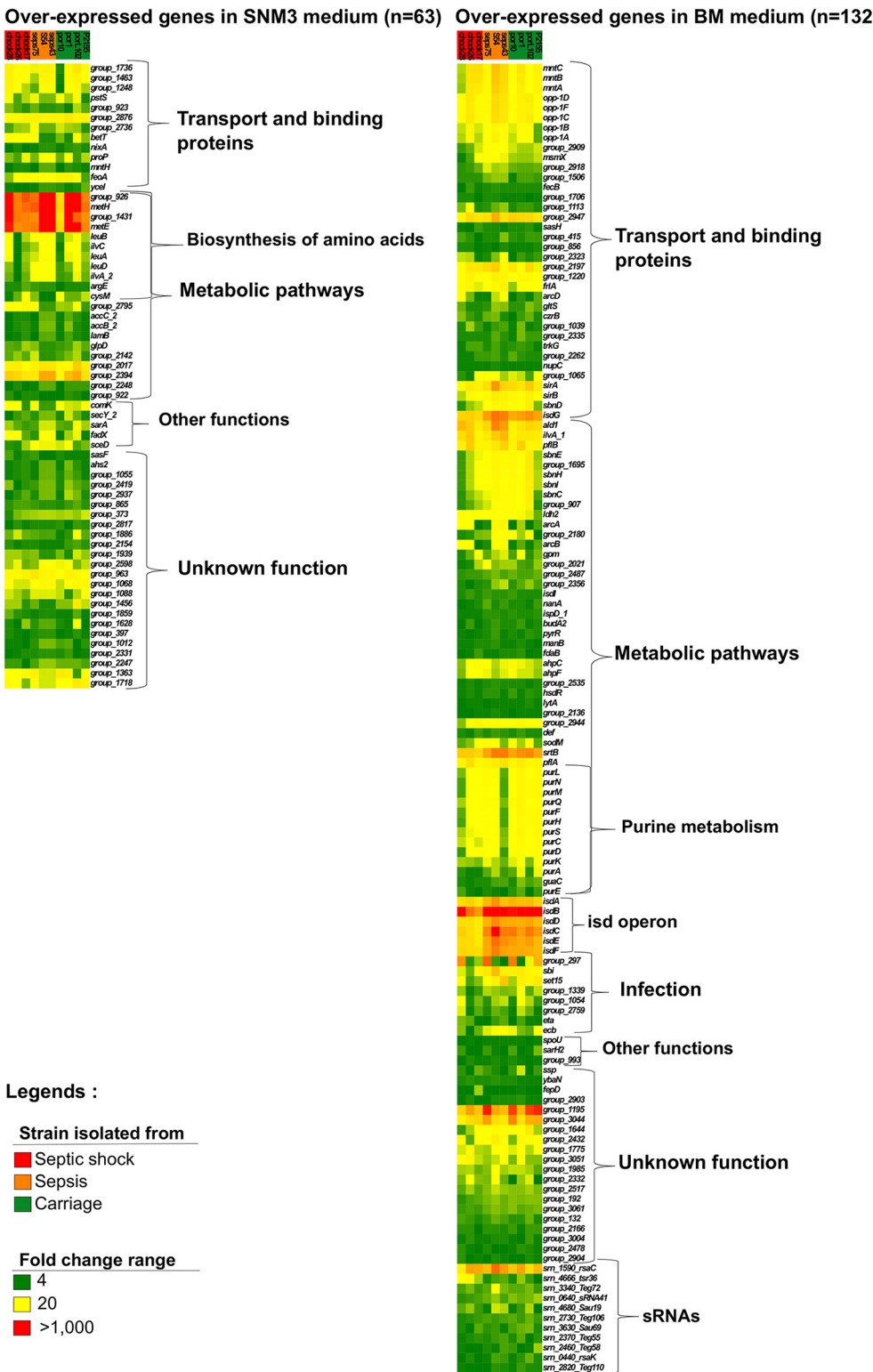

**FIG 6** Transcriptome profile of ST8 isolates under synthetic nasal conditions (SNM3) and bacteremia mimicking conditions (BMM). Heat maps display the intensity of genes overexpressed in SNM3 (left panel) and BMM (right panel). Strain names are indicated above each column, and their clinical metadata are indicated as follows: green, colonization; orange, sepsis; red, septic shock. The intensity range of overexpression within the heat maps is represented as follows: green, fold change up to 4; yellow, fold change up to 20; red, fold change of more than 1,000.

mimics some aspect of a bacteremia. Overall, 19.5% of genes (including sRNA genes) were differentially expressed between the two conditions, ranging from 15% for shock strain 28 to 22% for colonization strain P1 (Table S7). All of the septic shock strains presented an overall expression variation lower than that of the seven other strains. In BMM, the *isd* operon, involved in iron binding and transport, was overexpressed, with fold changes ranging from 200 to 1,000 as a function of the gene and strain considered (Fig. 6; Table S7). *sirA*, also involved in iron acquisition, was significantly induced (around 150 times). Among genes involved in virulence, *srtB* (41) was more than 300 times overexpressed in BMM while *sbi* and *spa*, encoding immune evasion factors, increased 200 and 80 times, respectively. Regarding sRNAs, a nearly 300-fold increase was observed for Srn_1590_RsaC, an sRNA that modulates the oxidative stress response during manganese starvation (42) and that is known to be overexpressed during acute *in vivo* infection (43). Apart from Srn_1590_RsaC, few sRNAs were strongly induced under these bacteremia-mimicking conditions. On the other hand, genes overexpressed in SNM3 were mostly involved in amino acid transport and metabolism, consistent with nutrient limitation in the human nose (44) (Fig. 6; Table S7). To summarize, based on the overall set of genes overexpressed in SNM3 and BMM, these data indicate that the *in vitro* culture media used in this study indeed mimic conditions that can occur during colonization and severe infections.

**(ii) Search for discriminant transcriptomic markers.** Based on the conditions used above, we searched for gene expression variations that could discriminate the clinical origin of the studied isolates. We pooled appropriate RNA-seq data in the categories of colonization, infections, bacteremia without organ failure, and septic shock. Additionally, we applied a DESeq cutoff of 2 and a *P* of <0.01 to identify the most significant variations (Table S7). First, we compared colonization and strains that yielded infections. In BMM, the expression of *norB*, encoding a quinolone resistance protein but also a participant in bacterial fitness within abscesses (36), was increased by more than 2-fold in infectious strains. In SNM3, the expression of *lrgA* and *lrgB* was lowered by around 3-fold in these strains, indicating that they could be used as biomarkers to forecast staphylococcal infections. Then, we compared strains isolated from healthy carriers with either bacteremia without organ failure or with septic shock strains. The comparison between bacteremia without organ failure and colonization showed that although no significant differences could be depicted in SNM3, the transcript levels of *radC* and *ald1* were modified in BMM by 0.4- and 2.4-fold, respectively (Table S8). More differences were detected between septic shock and colonization, with six and seven genes differentially expressed in BMM and SNM3, respectively. In BMM, all were mRNAs, either up- or downregulated, while in SMN3, one belonged to sRNAs (Srn_4470; i.e., SSR42, known to be involved in virulence [45–47]). Finally, we performed comparisons between strains isolated from bacteremia without organ failure and strains isolated from bacteremia with septic shock (Table S8). In BMM, 12 genes were differentially expressed. These included 10 mRNAs and 2 sRNAs. In SNM3, more differences could be observed, since 26 mRNAs and 3 sRNAs were differentially expressed. Among the mRNAs, the *isd* genes, encoding coagulase and several hemolysins, were upregulated in septic shock strains, indicating that some of the virulence equipment is already active during colonization-mimicking conditions. Additionally, study of the sRNAs revealed that RNAIII (Srn_3910) was also overexpressed in septic shock strains. Together, this transcriptomic analysis performed under nose-mimicking and blood infection-mimicking conditions showed that some genes are differentially expressed between the different categories and that septic shock strains exhibited higher transcript levels of a few virulence factors in SNM3. *lrgA* and *lrgB* are promising targets that could be used as primary tools to discriminate strains and, therefore, ultimately predict infections.

## DISCUSSION

*S. aureus* mortality is impacted by a variety of host-related factors, such as age and comorbidities (9), but the transition from colonization to infection, in both animals and humans, is much less understood. Also, the virulence of the pathogen has a significant

influence in the outcome of bacterial colonizations, which can lead to mild or severe infections. Combining genomic and transcriptomic approaches for pathogens might enhance our understanding of bacterial infections at the molecular level (48). Among bacterial pathogens, the identification of molecular signatures at DNA and/or RNA levels, which could prognosticate the issue of a colonization or an infection, would lead to the holy grail of clinicians, a step toward personalized medicine and infectious disease management.

Comparative genomic and transcriptomic analyses of multiple strains within a species could be a powerful means to uncover pathoadaptive genetic acquisitions and specific gene expression patterns. To date, several studies have focused on the genomic exploration of clinical isolates for both staphylococci and other bacterial pathogens. For instance, these include the identification of natural mutations associated with virulence (46), the evolutionary genomics study of host specificity (49), and the characterization of molecular signatures of persistent bacteremia in *S. aureus* (50). In this last study, comparative genomics revealed that isolates from persistent bacteremia have a low mutation frequency, but most are nonsilent mutations, suggesting that the *S. aureus* genome sequence can influence the clinical onset of infected hosts. Besides staphylococci, a recent study in *Streptococcus agalactiae* revealed the identification of CC-specific genes associated with virulence and conclude that this may explain differences in terms of virulence potential for certain CCs (51). Conversely, few studies performed whole-transcriptome comparisons between *S. aureus* strains to show that gene sets can be related to phenotypic features, such as bacterial invasion and host adaptation (52).

Here, we compiled a multicountry *S. aureus* genome data set containing 356 strains from livestock and humans with reliable clinical information responsible for colonization and nonsevere or severe infections. MLST representation of our *S. aureus* isolate data set indicates that our collection is a robust representation of the worldwide clonal diversity. Furthermore, our study fits with the reported prevalence of MRSA strains at a global scale (25, 53). The set of mobile genetic elements, some from phage origin, varies significantly between livestock and human isolates and also between colonization and infection strains, implying that analyzing the *S. aureus* mobilome gene content and expression is highly instructive. GWAS revealed that two genes, *sak* and *scn*, are absent in all livestock-specific strains but are detected in all strains infecting humans. Although promising, our findings also indicate that strains that belong to STs that can infect both animals and humans cannot be discriminated using those two markers. This result is congruent with previous findings indicating that all "*sak*-positive" strains are human associated (11). Staphylokinase (*sak*) is a virulence factor acting as a thrombolytic agent, which leads to tissue damage and improves bacterial invasiveness. The SAK protein has a dual role in human illnesses, promoting skin infections while, surprisingly, decreasing disease severity (54). The staphylococcal complement inhibitor SCIN (encoded by *scn*) helps the bacteria escape attack from the host immune system (55). The protein allows complement evasion by inhibiting the central complement convertases, reduces phagocytosis following opsonization, and binds to the alpha-defensins secreted from human neutrophils to counteract their bactericidal properties (56). Thus, SCIN is specific to human infections due to its ability to neutralize both the innate and adaptive immune host responses. Besides the SAK and SCIN genes, other determinants identified by Scoary and pyseer were phage-related functions and hypothetical proteins, suggesting the important role of mobile elements in *S. aureus* evolutionary history. Additionally, our analyses showed that sRNA genes, which contribute to tight gene expression regulation in bacteria (57), were not discriminant. This result was not surprising since several sRNAs are known to be implicated in staphylococcal virulence (58). Conversely, IGRs, which are often neglected in genomic analyses, provided significant information and consistency using both GWAS approaches, indicating that they might be useful.

However, in the present report, analysis of gene content and mutation rate showed some limitations in discriminating strains regarding the clinical status of their hosts and their origins. Therefore, we implemented RF, a well-established statistical method

that has two main advantages in our context: (i) RF can be used to rank the importance of the variables using permutations and out-of-bags, and (ii) RF is not limited to linear association and is flexible enough to catch nonlinear as well as interaction relationships between genes and infection status. Based on GWAS results, we explored CDS and IGR content but not the sRNA content. Again, the use of IGRs in combination or not with CDSs showed that these data are relevant to predict host specificity but can also be useful in increasing prediction of clinical status. Our out-of-bag prediction showed that RF was powerful in determining whether a strain was specific to humans or animals. Interestingly, determination of variables of importance indicated a good correlation between GWAS and RF. *sak* and *scn* were again within the short set of the most important variables along with some IGRs that were found significant using Scoary or pyseer (clusters 15862, 15817, 24735, and 16230). This suggests that a small set of genes may be sufficient to distinguish strains based on their genomic sequences. However, predictive assessment showed limitations as soon as the predictions went beyond host specificity. This lack of accuracy seems correlated with the size of the data set. In this retrospective work, a main limitation lies in the absence of a precise description of strain isolation conditions. From a set of thousands of *S. aureus* genome sequences available from the literature that we analyzed, only 356 met the necessary requirements (Fig. 1) for subsequent analyses. The close-up on strains from ST8 ($n = 80$) illustrates the importance of a large collection, as the error rate for predictions dramatically increased. An alternative hypothesis to explain more reliable predictions on the species level would be that strains belonging to a single ST (or specifically ST8) do not show strong differences in their propensity to cause invasive infection. However, from a positive standpoint, our data suggest that RF models should be implemented as additional information for clinician practitioners and that the collection of an extensive data set may lead to the development of dedicated ML programs based on the presence/absence of a defined set of genes.

Inadequacy of genome-based predictions performed on a limited number of isolates (i.e., the ST8 data set) was counterbalanced by RNA sequencing data. First, the use of SNM3 (44) and BMM (our study) revealed that these media can mimic colonization and bacteremia under laboratory conditions. A large set of genes associated with virulence and iron acquisition were overexpressed in blood, whereas *S. aureus* mostly expressed genes implicated in amino acid transport and metabolism in SNM3, consistent with the poor nutritional content of this medium mimicking human nose conditions. Among the virulence factors induced in BMM, the *isd* operon (59), immune evasion system genes *sbi* and *spa* (60), and *srtB*, which encodes a surface-anchored protein (61), are the most overexpressed genes. The comparison of strains based on their clinical descriptions allowed the identification of putative RNA markers with increased or decreased transcripts in subpopulations. Strains that led to infections had lower expression of the *lrgAB* operon in the nasal colonization medium SNM3, while *norB* increased under bacteremia conditions. The *lrgAB* operon encodes a murein hydrolase, with expression controlled by the LytSR two-component system (62), and the *norB* quinolone efflux pump (63) is known to participate in virulence in abscesses (36). Although the identification of *norB* can be interpreted as a bias due to quinolone treatment, an in-depth study of metadata from these strains, all isolated in our facility, indicated that the patients were not treated with this antibiotic. Therefore, this result should not be considered an artifact. Additionally, when we deciphered transcriptomes from strains that actually progressed to infections and compared infection severity, a second set of genes, larger than the previous one, was found significant and therefore could be used to forecast infection severity. To date, our study includes only 356 clinical isolates, but future broader investigations will be set to collect clones from asymptomatic carriers or from infected patients to build a larger collection to identify discriminant RNA markers.

To conclude, the use of comparative genomics, GWAS, RF analyses, and transcriptomics studies within a single pilot study identified genomic combinations that distinguish strain host specificity and enabled estimation, through ML algorithms, of whether a

colonized individual will be at higher risk to have a severe staphylococcal infection. We also highlight the importance of IGRs when considering the genomic elements. Furthermore, the expression pattern of a gene set may act as a biomarker to forecast strain clinical outcomes. The data from this study contribute to a better understanding of the critical factors that allow a pathogenic strain to establish a staphylococcal infection. In the future, a similar approach focusing on the role of host factors may be useful to provide an extensive prediction of risk associated with *S. aureus*.

## MATERIALS AND METHODS

**Construction of representative and harmonized *S. aureus* data sets.** Two different sets of genomes were generated during this study. The first one included strains that belong to multiple clonal complexes (see Table S1 in the supplemental material), whereas the second was devoted solely to ST8 isolates, which belong to CC8 (Table S1). The first set included genomes sequenced from strains of both animal and human origins. Most genomes were obtained from a literature survey. We used the terms "*Staphylococcus aureus*" AND "genomes" to search the PubMed database for studies published from July 2014 to June 2015, which corresponded with the beginning of the present study. Two authors (M. Sassi and Y. Augagneur) independently screened titles and abstracts to identify relevant studies. The search was supplemented by seeking the references of all eligible studies. Human studies were considered eligible when their authors (i) reported extractable and complete data on *S. aureus* genomes and (ii) documented sufficiently accurate clinical information. To prevent confounding factors, studies dealing with immunodeficient patients were excluded. A restriction on English and French literature was also imposed. To complete our data sets, we obtained strains from the infectious diseases and intensive care units located in our health care facility (Rennes University Hospital, France), from the Victoria Hospital (Australia), and from the Lausanne Hospital (Switzerland). Strains from asymptomatic individuals were also recovered from these facilities. Other strains were obtained from the National Reference Laboratory for Staphylococci in Lyon, France. All strains available in our laboratory were confirmed as being *S. aureus* by using multilocus sequence typing (64) and *S. aureus* protein A typing (Table S1). Then, two clinicians (P.-Y. Donnio and M. Revest) independently classified the clinical characteristics of patients from whom the bacterial strains were extracted. Patients were classified into three groups: those with severe infections, those with nonsevere infections, and those that were colonized. Strains for which both investigators could not agree were excluded from the analysis. Infections were considered severe if they led to bacteremia, infective endocarditis, pneumonia, or severe abscesses requiring surgery. Nonsevere infections were superficial infections treated on an outpatient basis. All other type of infections were excluded from the analysis. Patients with a superficial sample positive for *S. aureus* but without clinical symptoms were considered colonized. Additionally, *S. aureus* genomes from animal strains were kindly provided by R. Fitzgerald (Edinburgh, Scotland). The second set contained all the isolates typed as ST8 during the construction of the first set, to which we added 10 novel ST8 isolates for transcriptomic studies.

**Genome sequencing and assembly of additional *S. aureus* isolates.** *S. aureus* strains sequenced in Rennes facilities were grown in brain heart infusion (BHI) broth (Oxoid) at 37°C and under agitation (160 rpm). Genomic DNA was isolated using the Wizard genomic DNA purification kit (Promega) in the presence of lytic enzymes (lysozyme and lysostaphin) in accordance with the manufacturer's recommendations for Gram-positive bacteria. Subsequently, genomic DNA was precipitated with sodium acetate and washed two times with 70% (vol/vol) ethanol. DNA was sheared two times with a Covaris M220 focused ultrasonicator to generate an average fragment size of 600 bp. Unwanted smaller and larger fragments were removed by size selection using AMPure XP beads (Beckman Coulter). A DNA library was prepared using the NEBNext Ultra DNA library prep kit for Illumina (NEB) and sequenced as paired-end reads using an Illumina MiSeq platform and a MiSeq reagent kit v3 (600 cycles) (Illumina, Inc., San Diego, CA). Illumina reads for 27 strains (Table S1) were (i) trimmed using Trimmomatic (65), (ii) filtered based on quality using the Fastx-toolkit (http://hannonlab.cshl.edu/fastx_toolkit/) and, (iii) assembled using the SPAdes software (66, 67). SIS and GapFiller version 1.10 (68, 69) were used to improve the initial set of contigs, and the closest complete genome was used as the reference to order and orient the contigs.

**_S. aureus_ genomic annotation and pangenomic analyses.** To normalize coding sequence predictions, genomes were annotated using Prokka (70). Genes from each strain were functionally annotated using COG scripts (https://github.com/transcript/COG) and the COG databases (31). The number of genes encoding each COG family function were used to investigate pangenome associations with host and clinical phenotypes. We used R software to calculate the mean, the standard deviation, and $P$ values ($t$ test). The difference between COG families was considered significant when $P$ was <0.01. Figures were generated using ggplot package in R software. sRNA genes were predicted after extracting all sRNA sequences available in the SRD database (27) (http://srd.genouest.org/) and were subsequently aligned with each genome using BLAST and a homemade script (https://github.com/mosassi/Staphylococcus_annotation.git). The pangenome is the sum of the core genome (the set of genes present in all genomes) and the dispensable, accessory genome containing genes present in some but not all the strains, as well as strain-specific genes (71). The pangenomic analysis was performed on CDSs, sRNA genes, and intergenic regions (IGRs). Gene homologies between strains were assessed using Panaroo (26) with a >95% nucleotide identity cutoff and a refinding step with at least 50% of the sequence within a radius of 1,000 nucleotides to identify genes that were missed during annotation. The sRNA homologies between the strains were evaluated using BLAST and with >90%

nucleotide identity and >70% sequence coverage. The IGR homologies between the strains were estimated using Piggy software (72) with >80% nucleotide identity and >60% sequence coverage cutoffs. Core genome alignments were performed with PRANK (73) and with single nucleotide polymorphisms (SNPs) extracted. Antibiotic resistance and virulence genes were annotated using BLAST, ResFinder (74), and VirulenceFinder (75) databases.

**Minimum spanning trees and phylogenetic reconstructions.** Multilocus sequence typing (MLST) was performed on all genomes, and the results are presented in Table S1. In order to define the relationships among strains at the microevolutionary level, we performed allelic profile-based comparisons using a minimum spanning tree (MST) analysis with BioNumerics v7.6 software (Applied-Maths, Sint-Martens-Latem, Belgium). MST analysis links profiles so that the sum of the distances (number of distinct alleles between two STs) is minimized. Strains were grouped into clonal complexes (clonal families), defined as groups of profiles differing by no more than one gene from at least one other profile of the group. Accordingly, singletons were defined as STs having at least two allelic mismatches with all other STs.

A phylogenetic tree was constructed by considering the 26,577 polymorphic sites retained in the core genome. A transversion substitution model was selected on the basis of the Akaike's information criterion with jModelTest2 (76). Maximum likelihood phylogeny was constructed using PhyML with 1,000 nonparametric bootstrap iterations (77). A phylogenetic tree was visualized using Figtree (http://tree.bio.ed.ac.uk/software/figtree/).

**Selection signatures.** To infer homoplastic SNPs that might be driven by convergent and positive selection, we analyzed the global SNP matrix with the HomoplasyFinder algorithm (33).

**GWAS.** Two genome-wide association study (GWAS) analyses (Scoary and pyseer) were performed to investigate pangenome associations with host and clinical phenotypes. The input for both approaches were CDS and IGR presence/absence and SNP matrices (78). Using Scoary, genetic elements that matched our inclusion criteria (Bonferroni's $P < 10^{-4}$; best pairwise comparison $P$ value of <0.01) were considered significant. Due to the strong population structure of our complete collection and the dominance of the ST8 type, an in-depth analysis of genetic elements associated with a host or with a severity outcome was also performed using pyseer (v.1.3.6) (79). To do this, a phylogenetic distance file was calculated by using the phylogeny_distance.py included in pyseer on the maximum likelihood phylogeny. The distance, trait, gene, IGR, and sRNA presence and absence matrices and SNP files were used to run pyseer as described in the pyseer tutorial (https://pyseer.readthedocs.io/en/master/tutorial.html). A genome-wide association was considered statistically significant if the likelihood ratio test (lrt) $P$ was <0.00001.

**RF.** To build a classifier of the infection status of patients with respect to their genome, we used the well-known ensemble learning method called random forest (RF) (80). RF aims at building a forest of individual decision trees (obtained by applying the classification and regression tree algorithm), combined with randomized node optimization and bagging. We applied a random forest algorithm to our data set by using the software R (81) (function randomForest from the package randomForest maintained by A. Liaw and M. Wiener [82]). We used a total of 500 trees for each forest ($n_{tree} = 500$) and a number of variables randomly sampled at each split given by $srqt(p)$, where $p$ is the number of explanatory variables in a matrix of predictors ($mtry = sqrt(p)$).

**Transcriptomic analysis. (i) Growth and RNA extraction.** Only ST8 colonization isolates and ST8 strains recovered from bacteremia were used at this stage. *S. aureus* was precultured in tryptone soya broth (TSB; Oxoid) diluted 5 times in RPMI 1640 (Life Technologies) under agitation at 37°C. Overnight cultures were washed twice in synthetic nasal medium SNM3 (44) and then transferred in fresh SNM3 medium at an optical density at 600 nm ($OD_{600}$) of 0.1. Cells were grown overnight and monitored by measuring the $OD_{600}$. Appropriate volumes of cells were centrifuged and transferred either to fresh SNM3 or to fresh bacteremia-mimicking medium (BMM) at an $OD_{600}$ of 0.1. BMM was prepared from 50% human AB serum (Etablissement Français du Sang), 44% RPMI 1640, and 6% packed red blood cells (washed in RPMI 1640 and set to a hematocrit of 50). Then, *S. aureus* was incubated for 4 h at 37°C under agitation. Cells were harvested by centrifugation at 4,500 rpm for 8 min, and pellets were washed with 5 mL of $H_2O$, allowing lysis of red blood cells, and centrifuged again at 4,500 rpm for 8 min, and pellets were frozen prior to RNA extraction. Dried frozen pellets were resuspended in 500 $\mu$L of cold lysis buffer (20 mM sodium acetate, 1 mM EDTA, 0.5% SDS, pH 5.5). Total RNAs were extracted by phenol (pH 4) using a FP120 FastPrep cell disruptor (MP Biomedicals), as described previously (83).

**(ii) cDNA library synthesis and Illumina RNA sequencing.** Up to 10 $\mu$g of total RNA was doubled treated with amplification-grade DNase I (Invitrogen) to remove genomic contaminations. The absence of DNA was checked by quantitative PCR (qPCR) with primers targeting *S. aureus hup* in an Applied Biosystems 7500 instrument. RNA integrity was verified on a bioanalyzer (Agilent). rRNAs were depleted using a Ribo-Zero magnetic kit (Epicentre), and the remaining RNAs were purified by ethanol precipitation according to the manufacturer's recommendations. Efficiency of depletion was estimated on a bioanalyzer (Agilent). Stranded cDNA libraries were prepared using the NEBNext Ultra directional RNA library prep kit for Illumina (New England Biolabs) following an initial fragmentation step of 13 min at 95°C resulting in cDNA libraries of ~330 bp. The concentration, quality, and purity of the libraries were determined using a bioanalyzer, a Qubit fluorometer (Invitrogen), and a NanoDrop spectrophotometer (Thermo Scientific). Indexed libraries were equimolarly mixed in two pools and sequenced on an Illumina HiSeq 1500 system (high output, 200 cycles, paired end), as described in the manufacturer's instructions.

**(iii) Read mapping and differential expression analysis.** All genomes from strains used for transcriptome analysis were annotated using Prokka (70). Genes from each strain were functionally annotated using the KEGG database (84). Two annotation files (in GFF format) were generated per

strain. One contained all CDSs predicted from Prokka, and the other was composed of all sRNA genes identified based on sequence similarity with the SRD database (27). To allow comparisons between all strains sequenced, orthologous genes were clustered by sequence homologies and unique identifiers were generated based on pangenomic analyses as described above. Quality control of RNA-seq reads was done as described previously (27). Illumina reads were trimmed using the FASTX-Toolkit (http://hannonlab.cshl.edu/fastx_toolkit/) and then mapped onto their respective genomes. SAM files were filtered on bitwise flag values (85) to keep only properly paired reads and counted by HTSeq count (86) for stranded library with the intersection nonempty mode. Fragments per kilobase per million (FPKM) normalization was calculated to remove weakly expressed transcripts. To do so, we removed all transcripts which led to an FPKM lower than 10 in each strain and condition tested. Then, differential expression analyses were calculated using DESeq with the per-condition method and the parametric fit type (threshold $P = 0.05$; fold change threshold = 4; baseMean $> 15$) (87) for SNM3 medium versus BMM or with the pooled method and the parametric fit type (threshold $P = 0.01$; fold change threshold = 2; baseMean $> 15$) to search for transcriptomic markers.

**Data availability.** The genomes sequenced during this study are available under the following BioProject accession numbers: PRJNA273632, PRJNA280933, and PRJNA290551. The RNA sequences have been submitted to the NCBI SRA database under BioProject accession number PRJNA647528.

## SUPPLEMENTAL MATERIAL

Supplemental material is available online only.

**FIG S1**, PDF file, 0.4 MB.
**FIG S2**, PDF file, 0.1 MB.
**TABLE S1**, XLSX file, 0.04 MB.
**TABLE S2**, XLSX file, 10.5 MB.
**TABLE S3**, XLSX file, 0.04 MB.
**TABLE S4**, XLSX file, 0.2 MB.
**TABLE S5**, XLSX file, 13.6 MB.
**TABLE S6**, XLSX file, 0.1 MB.
**TABLE S7**, XLSX file, 0.01 MB.
**TABLE S8**, XLSX file, 0.2 MB.

## ACKNOWLEDGMENTS

We thank Ben Howden and Ross Fitzgerald for kindly providing human and animal strains. We also thank the Roscoff Bioinformatics platform ABiMS (http://abims.sb-roscoff.fr) for providing computational resources and the BioGenouest GEH facility (https://geh.univ-rennes1.fr/) for technical support.

We declare that we have no competing interests.

This work was supported by Inserm, University of Rennes1, and BIOSIT. M.S. was supported by Region Bretagne grant no. SAD SARS 8254 and SARS_2 9181 (to Y.A.). J.B. was a recipient of a fellowship from the Direction Générale pour l'Armement (Agence Innovation Défense) and the Conseil Régional de Bretagne.

The funders had no role in study design, data collection and interpretation, or the decision to submit the work for publication.

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
