## [Reviewer comments · mSystems]

Forecasting *Staphylococcus aureus* infections using genome-wide association studies, machine learning and transcriptomic approaches

Mohamed Sassi, Julie Bronsard, Gaetan Pascreau, Mathieu Emily, Pierre-Yves Donnio, Matthieu Revest, Brice FELDEN, Thierry Wirth, and Yoann AUGAGNEUR

Corresponding Author(s): Yoann AUGAGNEUR, Universite Rennes 1 Inserm U1230

Review Timeline:

Submission Date:	April 25, 2022
Editorial Decision:	May 31, 2022
Revision Received:	June 7, 2022
Accepted:	June 8, 2022

Editor: Benjamin Wolfe

Reviewer(s): Disclosure of reviewer identity is with reference to reviewer comments included in decision letter(s). The following individuals involved in review of your submission have agreed to reveal their identity: Nicole Wheeler (Reviewer #3)

Transaction Report:

DOI: <https://doi.org/10.1128/msystems.00378-22>

May 31, 2022

Dr. Yoann AUGAGNEUR
Universite Rennes 1 Inserm U1230
Inserm U1230 BRM
2 Avenue du Pr Leon Bernard
RENNES
France

Re: mSystems00378-22 (Forecasting *Staphylococcus aureus* infections using genome-wide association studies, machine learning and transcriptomic approaches)

Dear Dr. Yoann AUGAGNEUR:

Thank you for submitting your manuscript to mSystems. We have completed our review and I am pleased to inform you that, in principle, we expect to accept it for publication in mSystems. However, the reviewers have noted some minor changes/modifications that are needed to improve clarity and grammar in a few instances (see comments from Reviewers at the bottom of this email).

Below you will find instructions from the mSystems editorial office and comments generated during the review.

Preparing Revision Guidelines

Sincerely,

Benjamin Wolfe

Editor, mSystems

Journals Department
American Society for Microbiology
1752 N St., NW

Reviewer comments:

Reviewer #1 (Comments for the Author):

This resubmission by Sassi et al. highlights the global diversity of *Staphylococcus aureus* (SA). The major goal of this work was to identify specific markers of SA strains that cause severe human infection and to differentiate those strains from animal isolates or human isolates that do not cause invasive disease. Their analysis emphasizes the impressive genomic differences between the numerous SA clonal complexes (CCs). Some of the highlights of their findings include that 78% of the SA pangenome is accessory, there is substantial diversity of predicted small RNAs (sRNA) as well as in non-coding intergenic regions (IGRs), and clear genetic markers that discriminate between isolation source (animal vs human) even though all isolates were distributed across the SA phylogeny independent of isolation source. The authors added random forest (RF) classification algorithms to their analysis and describe the power of RF to differentiate isolation source. The authors observed that methods to identify genetic markers specific for SA isolates from invasive human disease were less successful compared to analysis by isolation source. Moreover, these markers were less successful at predicting whether a strain would be associated with invasive disease relative to human carriage. This was also true within a genetically related single clonal complex. To provide additional discriminatory power, the authors analyzed the transcriptome of carriage and disease isolates within a single clonal complex when shifting from SNM3 (mimic nasal colonization) to BMM (mimic bacteremia) media. The authors observed that shifting from SNM3 to BMM resulted in a significant change in the transcriptome (just under 20% of transcripts were altered). They then interrogated this data to determine any transcriptomic markers of carriage and disease isolates. They identified *norB*, involved in quinolone resistance, to be significantly upregulated by disease isolates in BMM and the *IgrAB* genes, encoding a murine hydrolase, to be significantly downregulated by disease isolates in SNM3 compared to carriage isolates.

Upon resubmission the authors have adequately addressed the concerns I had from their initial submission. I have no further concerns.

Reviewer #3 (Comments for the Author):

The authors have conducted a substantial revision of their paper, using a more accurate pangenome analysis tool, employing a more sophisticated machine learning approach, and adding an analysis of noncoding DNA. These changes add value to the manuscript, and it is now at a point where I only have minor comments to make.

There are spelling and grammar issues with the revised manuscript that should be resolved before publication.

Line 107/353: an alternative explanation for why prediction works on the species level but not the CC level is that individual clonal complexes, or CC8 specifically, don't show a strong difference in inter-strain propensity to cause invasive infection. We know that host factors play a role in infection outcome, so there will be some component of infection outcome that can't be predicted based on data on the bacteria.

Line 145: how were MRSA strains identified? Is this phenotypic data or inferred from the genome?

Line 163: it's not clear what the authors mean by sRNA genes are more widely distributed in the core genome than protein coding genes - what does widely distributed mean?

Random forest results: please plot predicted and actual phenotypes against a phylogenetic tree so we can see whether errors cluster in particular clades or appear random.

Line 384: the results of this study do not confirm that the culture conditions reflect colonization and infection conditions, as a direct comparison between these lab approaches and the conditions they are trying to mimic isn't performed in this study.

Table 1: please name the genes flanking each of the significant IGRs for biological context.

Supplementary Table 3: please do the same here, to save the reader needing to fetch all of the information from Supp Table 5

June 8, 2022

Dr. Yoann AUGAGNEUR
Universite Rennes 1 Inserm U1230
Inserm U1230 BRM
2 Avenue du Pr Leon Bernard
RENNES
France

Re: mSystems00378-22R1 (Forecasting *Staphylococcus aureus* infections using genome-wide association studies, machine learning and transcriptomic approaches)

Dear Dr. Yoann AUGAGNEUR:

I am pleased to inform you that your manuscript has been accepted, and I am forwarding it to the ASM Journals Department for publication.

Before it can be scheduled for publication, your manuscript will be checked by the mSystems production staff to make sure that all elements meet the technical requirements for publication. They will contact you if anything needs to be revised before copyediting and production can begin. Otherwise, you will be notified when your proofs are ready to be viewed.

Publication Fees:

We recognize that the video files can become quite large, and so to avoid quality loss ASM suggests sending the video file via <https://www.wetransfer.com/>. When you have a final version of the video and the still ready to share, please send it to mSystems staff at mSystems@asmusa.org.

For mSystems research articles, if you would like to submit an image for consideration as the Featured Image for an issue, please contact mSystems staff at mSystems@asmusa.org.

Sincerely,

Benjamin Wolfe
Editor, mSystems

Journals Department
Table S4: Accept

Table S7: Accept

Table S6: Accept

Table S8: Accept

Supplemental Figure S1: Accept

Supplemental Figure S2: Accept

Table S3: Accept

Table S1: Accept

Table S2: Accept

Table S5: Accept